# Assessment of the Nutritional Quality of Plant Lipids Using Atherogenicity and Thrombogenicity Indices

**DOI:** 10.3390/nu14183795

**Published:** 2022-09-14

**Authors:** Sarvenaz Khalili Tilami, Lenka Kouřimská

**Affiliations:** Department of Microbiology, Nutrition, and Dietetics, Czech University of Life Sciences Prague, Kamýcká 129, 165 00 Prague, Czech Republic

**Keywords:** fatty acids, atherogenicity index, thrombogenicity index, cholesterol, plant lipids

## Abstract

Dietary lipids derived from plants have different compositions of individual fatty acids (FA), providing different physical and chemical properties with positive or adverse health effects on humans. To evaluate the nutritional value and assess the FA composition of various plants, the atherogenicity (AI) and thrombogenicity (TI) indices were calculated and reviewed for nine different categories of fats and oils. This included common oils, unconventional oils, nut oils originating from temperate regions, Amazonian and tropical fats and oils, chia seed oil, traditional nuts originating from temperate regions, unconventional nuts, seeds, and fruits, and their products. The main factors influencing fatty acid composition in plants are growth location, genotype, and environmental variation, particularly temperature after flowering, humidity, and frequency of rainfall (exceeding cultivar variation). The lowest AI was calculated for rapeseed oil (0.05), whereas the highest value was obtained for tucuman seeds (16.29). Chia seed oil had the lowest TI (0.04), and murumuru butter had the highest (6.69). The differences in FA composition and subsequent changes in the lipid health indices of the investigated fats and oils indicate their importance in the human diet.

## 1. Introduction

Oils obtained from the seeds, fleshy parts of the fruits of different plants (e.g., olive and palm), legumes, and nuts are utilized in various sectors [1], such as cosmetic and personal care products, supplements, and manufactured products, in addition to the culinary and food industry. Dietary lipids are an important source of energy for living animals. They can provide high amounts of essential polyunsaturated fatty acids (PUFAs), particularly *n*-3, which have positive health effects for preventing several diseases, such as reducing the risk of coronary heart disease and strokes, inflammation, and several cancers, and are important for the development and functionality of the brain and retina [2,3]. In addition, they are good sources of micronutrients and bioactive components that are important for the development and function of different organs [2]. Various edible oils with different physicochemical attributes are available. Some varieties are used less and are considered unconventional oils, alongside nut oils, Amazonian and tropical fat and oils. Differences in the fatty acid composition of oils result in differences in their functionality and properties.

From the nutritional point of view, the proportion of specific groups of FAs has special importance [4]. In today’s Western diet, the excessive intake of *n*-6 PUFAs results in a very high ratio of *n*-6/*n*-3, which can promote the pathogenesis of several diseases, such as cardiovascular diseases, diabetes, depression, immune disorders, and neurological dysfunction [4,5,6]. However, the *n*-3/*n*-6 ratio in fish and seafood is often discussed as the opposite ratio results in very low values. The role of most of the main classes of unsaturated fatty acids (UFAs) in inhibiting plaque accumulation and reducing the levels of phospholipids and cholesterol has been well documented [7]. Increased attention needs to be paid to *n*-3 PUFAs since most diets are high in *n*-6 PUFAs. Limited amounts of *n*-3 dietary sources are found in vegetable oils, vegetables, dairy products, walnuts, and flaxseeds. Fish and seafood, especially fatty fish, are excellent sources of eicosapentaenoic (EPA) and docosahexaenoic (DHA) fatty acids [2,5,8]. It has been well documented that it reduces the risk of coronary heart disease and increases low-density lipoprotein (LDL) cholesterol in humans, replacing excess saturated fatty acids (SFA) in the diet with PUFAs [9,10], even recommending *n*-6 PUFA replacement [11]. The optimal ratio for *n*-6 to *n*-3 is recommended to be 1:1 to 4:1 [5]. According to the EFSA [11], no specific values for the *n*-3/*n*-6 ratio was set. There is the recommendation for the intake of *n*-3 PUFAs mainly EPA and DHA for adults to be at least 250 mg per day and during pregnancy and lactation, in addition to the above-mentioned intake, 100–200 mg performed DHA have been recommended. For infants less than 6 months and young children up to 24 months, adequate intake of 100 mg DHA have been recommended [11]. According to the EFSA report, the intake of 2 g α-linolenic acid (ALA) for general population has been recommended [11]. *N-*6 PUFA mainly consists of linoleic acid (LA) and to a lesser amount arachidonic acid (ARA). The intake of 10 g of *n*-6 LA per day for adults has been recommended.

## 2. Nutritional Quality Indices

Assessing the indices of atherogenicity (AI) and thrombogenicity (TI) can provide information regarding the different effects that individual FAs could have on human health and, in particular, on the likelihood of an increased incidence of atherosclerosis, the development of blood clots, atheroma, and thrombus formation [12,13]. AI indicates the relationship between major FAs (lauric, myristic, and palmitic) classified as proatherogenic and antiatherogenic UFAs [13]. The incidence of atherogenicity has been associated with various inflammatory and innate immune pathways, and the atherogenic index may be used as a preliminary indication of accelerated atherosclerosis associated with numerous inflammatory pathways [14]. TI shows a tendency for clots to form in blood vessels. This process is defined as the relationship between pro- thrombogenic (SFA) and anti-thrombogenic fatty acids (monounsaturated fatty acids (MUFA), *n*-6, and *n*-3 PUFA) [13]. Other indices can evaluate the lipid quality in various types of fats and oils, including HH (hypocholesterolemic-hypercholesterolemic ratio), HPI (health-promoting index), UI (unsaturation index), FLQ (fish lipid quality/flesh lipid quality), LA/ALA (linolenic acid/α-linolenic acid ratio), and TFA (trans fatty acid) [7]. AI and TI are one of the most well-known, reliable, and widely used indices for all kinds of fats and oils that can be used to assess the potential effects of FA on cardiovascular disease [7]. The atherogenic index shows a precursory indication of accelerated atherosclerosis and can support our knowledge of the numerous inflammatory pathways associated with it, while the thrombogenic index shows a tendency for clot formation in the blood vessels and cardiovascular disease [12,15].

This study aimed to compare the traditional (commonly used oil or conventional) oils and nuts with non-traditional (less used, non-conventional) plant sources of lipids to evaluate them from a nutritional point of view concerning human health. In line with the food industry’s interest in creating more valuable products from seed oil or unconventional oils (e.g., borage oil and pumpkin oil) [16] and due to their various health-promoting attributes, this type of edible oil has gained increasing attention. The objective of the present study was to develop literature on the assessment of the nutritional quality of plant lipids and their potential for the prevention of several diseases using indices of atherogenicity (AI) and thrombogenicity (TI). The results of this study were collected based on the literature searches performed on the FA profile as well as SFA, MUFA, PUFA, *n*-6 PUFA, *n*-3 PUFA, *n*-6 PUFA/*n*-3 PUFA. AI, and TI. Health indices of atherogenicity and thrombogenicity in dietary sources were determined based on the data obtained from the fatty acid composition of plants, and their importance to human food was evaluated.

Furthermore, the locality and climatic region could affect the composition of individual FAs and thus the indices of atherogenicity and thrombogenicity of individual plant lipids. Therefore, conventional plant lipid sources were compared with unconventional sources based on their origins in the current study.

## 3. Material and Methods

In this study, AI and TI were calculated based on reviewed and screened articles. The data were categorized into different oil types (common oils, unconventional oils, nut oils, Amazonian fats and oils, tropical fats and oils, chia seed oil, traditional nuts, unconventional nuts, seeds and fruits, nuts, and seed products), and sorted according to the atherogenicity index from the lowest value to the highest. Some nuts and oils could belong to multiple groups. However, the classification was based on the similarity of the FA composition within the existing category (e.g., as a tropical oil, henna oil was presented in the unconventional oils from the temperate region due to the similarity of FA composition with the other oils; and peanut oil, which belongs to the tropical and semi-arid tropical region, was presented in the nut oils with the temperate origin).

The indices of atherogenicity (IA) and thrombogenicity (IT) were calculated according to Ulbricht and Southgate [12] using the following equations:$$\mathrm{IA}=\frac{C12:0+4\times C14:0+C16:0}{\Sigma \mathrm{MUFA}+\;\Sigma \mathrm{PUFA}\;n-6+\;\Sigma \mathrm{PUFA}\;n-3}$$
$$\mathrm{IT}=\frac{C14:0+C16:0+C18:0}{0.5\;\Sigma \mathrm{MUFA}+0.5\;\Sigma \mathrm{PUFA}\;n-6+3\;\Sigma \mathrm{PUFA}\;n-3+\frac{\Sigma \mathrm{PUFA}\;n-3}{\Sigma \mathrm{PUFA}\;n-6}}$$

Cluster analysis was performed using STATISTICA v.12.0 (StatSoft, Inc., Tulsa, OK, USA). Multidimensional cluster analyses were applied to selected groups in each category to show the differences between samples from individual groups, including SFA, MUFA, PUFA, *n*-3, and *n*-6 PUFA, the ratio of *n*-3/*n*-6 PUFA, AI, and TI. SPSS software (version 16, Chicago, IL, USA) was used to analyze the data, and Levene’s test was applied for normality and homogeneity of variance. The effects of different oil types on the AI and TI indices were examined using one-way analysis of variance (ANOVA). Differences between several groups were determined using Duncan’s post hoc test.

## 4. Results and Discussion

### 4.1. Plant Oils

Based on the collected data of the individual FAs in plant lipid sources obtained from the available literature, the sum of FAs, the ratios of *n*-3/*n*-6 PUFAs, values of AI and TI of the common and unconventional and non-traditional (e.g., Amazonian and tropical oil, and nut oil.), were calculated (Table 1). The sources of the data for the FA composition of each oil are listed in Table 2. The lowest atherogenic index was observed for rapeseed oil (AI = 0.05), whereas the highest value was observed for olive oil (AI = 0.16). Soybean and sesame oils had the same value (AI = 0.11). IA and IT values are close to the values stated for the so-called Eskimo diet, which indicates a very low incidence of coronary heart disease (IA, 0.39; IT, 0.28) [12]. Differences are shown in the hierarchical cluster graphs (Figure 1, Figure 2, Figure 3, Figure 4, Figure 5, Figure 6 and Figure 7).

In the category of tropical oils, avocado oil has gained increased attention in the human diet mainly as a result of the favorable amount of MUFA (65.3%) [30] and positive health effects regarding anti-inflammatory properties and cardiovascular disease [35]. The content of 18:3 (*n*-3), the most important PUFA, in avocado oil and olive oil was almost the same (0.73 and 0.78%, respectively). Regarding the cardiovascular health indices, the same IT value (0.40) was observed for both oils. The benefits of consuming oils rich in oleic acid (18:1 *n*-9), including olive oil (approximately 75%), avocado oil (60.61%), rapeseed oil (60.30%), and sunflower oil (53.11%) [17], have been associated with lowering blood cholesterol and preventing the development of atherosclerosis [36]. Among plant oil crops with high oil content, sesame oil [37] is a fatty acid source with similar MUFA and PUFA contents. Walnut and corn oil have a higher PUFA content [19], and olive oil has a higher MUFA [38,39]. The highest *n*-6 content as a precursor of essential fatty acids, including linoleic acid (18:2 *n*-6; 72.5%) [18] and γ-linolenic acid (18:3 *n*-6), was observed in evening primrose (*Oenothera biennis*) [40], which is associated with pro-inflammatory properties and might have adverse health effects in the case of excessive consumption. Linseed, rapeseed, and soybean oil had the highest amounts of 18:3 *n*-3 (55.20, 10.40, and 8.03%, respectively).

Pritchard et al. [20] showed that the total oil content was generally higher in rapeseed grown in wetter and colder regions, which provides longer maturation for the crop, leading to a longer period for oil accumulation. A low oil content correlated with higher temperatures during spring, flowering, and seed ripening. The importance of post-flowering temperature in the FA composition of many oilseeds has been well documented [20]. The lower the post-flowering temperature, the higher the degree of oil unsaturation. The influence of temperature and rainfall (higher oil content was observed in the seeds from years of high spring rainfall) on the oil composition was bigger than the influence of the region, in which the rapeseed was grown. In the case of peanuts, low temperatures are associated with lower saturation [41]. It was also demonstrated that by increasing the temperature in the early stage of the grain-filling period, the oleic/linoleic acid content increases [42,43]. Based on reported studies focusing on determining the geographic origin of virgin olive oils using chromatographic methods and infrared spectroscopy, geographic variation is reflected by differences in the fatty acid composition of plants [44]. For example, coconut oil consists of the medium-chain SFA lauric acid (C12:0), and tropical oils are generally solid fats with a high resistance to oxidation at high temperatures, whereas olive oil obtained from the Mediterranean Basin is liquid.

### 4.2. Lesser-Used and Non-Traditional Oils (Unconventional Oils) Originating from the Temperate Region

Among the reviewed studies of the lesser-used and unconventional oils, the lowest AI was found in hemp oil from Turkey (AI = 0.07). Kiralan et al. [21] reported that oils from non-drug varieties of cannabis seeds (hemp oil) in northwestern Turkey had high amounts of PUFAs (89.43–90.63%), including linoleic acid as the most representative (55.41–56.94%), followed by α-linolenic acid (16.51–20.40%) and oleic acid (11.40–15.88%). Studies have further shown that cannabis seeds from temperate or warm climates contain low amounts of γ-linolenic acid, whereas higher amounts of γ-linolenic acid are observed in temperate or cold regions [21]. A high amount of PUFA (˃50% of the total mass), as well as the low levels of AI of pumpkin and field pumpkin (squash) oil (0.21 and 0.20%, respectively) [23], confirms their good nutritional quality. However, higher TI values were reported for pumpkin and field pumpkin oil (0.47 and 0.56). Linoleic acid (18:2 *n*-6) and oleic acid (18:1 *n*-9) were the major unsaturated fatty acids in both borage and pumpkin oil, representing the (38.30, 54.70, and 50.88%) and (17.95, 24.52, and 25.82%), respectively [17,18,23]. Cacao seeds cupuassu had three major FAs including oleic acid (18:1 *n*-9), stearic acid (18:0), and α-linolenic acid (42.30, 35, and 11%) [31]. Of all the investigated common oils, evening primrose (*Oenothera biennis*) oil is a good source of linoleic acid (LA, 18:2 *n*-6) (72.5%) and γ-linolenic acid (GLA, 18:3 *n*-6) [40]. Blackcurrant oil has a unique fatty acid profile and is a valuable source of a wide range of bioactive compounds and antioxidants that can be used in the pharmaceutical industry. Fruit and berry oils are characterized by gentle processing (e.g., no refining and cold pressing), unique aroma, health-promoting attributes, low production yields, and high prices. It was demonstrated that despite the low content of 18:3 *n*-3 FA in rice bran oil, the amount is sufficient for biosynthesizing *n*-3 FAs, particularly EPA and DHA, resulting in prolonged shelf life compared to the common oil [45]. Wheat germ oil had the highest atherogenic index at 0.29. Field pumpkin had the highest TI at 0.56. Identical AI values of 0.01 were obtained for blackcurrant and henna oils. Considering the similarity of FA composition and its lesser usage, henna oil, which originated from the tropical and subtropical areas [22], was placed in this category. It has the highest PUFA content (particularly *n*-6 PUFA) and can be used for numerous medicinal purposes, including as a hair treatment; to treat eczema, hypotension, bleeding, and cervicitis; and has astringent, cardioinhibitory properties [46].

### 4.3. Nut Oils, Nuts, Seeds, and Their Products

Nut oils contain high amounts of unsaturated fatty acids, which result in favorable indices of atherogenicity and thrombogenicity; however, they are prone to oxidation. In the investigated nut oil group, the lowest AI was found for pecan and almond oils (AI = 0.07), a low risk of cardiovascular disease. Pistachio oil showed a higher AI compared to the other nut oils (AI = 0.15), indicating a medium risk of cardiovascular disease. In general, owing to the low AI value, all categories of nut oils have beneficial health effects. Compared with other nut oils, walnut oil has a high PUFA content (60.40%). Although they are a good source of *n*-3 FAs, they cannot be considered as cooking and frying oils owing to their low oxidative stability. Generally, nut oils have higher MUFA than PUFA and SFA, except for the above-mentioned oils. Additionally, peanut and pistachio oils have high oxidative stability.

Commonly consumed nuts have a low SFA content, a high UFA content, particularly MUFAs, and are the rich sources of *n*-3 FAs (18:3 *n*-3). Their composition is unique as, on average, they have less SFA than olive oil and slightly more SFA than rapeseed and sunflower oil. Nuts have similar oleic content to rapeseed oil but a lower amount compared to olive oil [27]. Nuts and rapeseed oil showed the same linoleic content (18:2 *n*-6), significantly higher than that of olive oil. Traditional nuts show low indices of AI (0.07–0.14) and TI (0.16–0.35). In the unconventional nuts and seeds group, the lowest AI was found for pecans (AI = 0.12), whereas the highest was found for tucuman seeds (AI = 16.29).

Alves et al. [32] stated that baru almonds and cashew nuts from the Cerrado region of South America are good sources of high-quality plant lipids as they have a high ratio of MUFA to SFA (3.33 and 2.75%, respectively). Oleic acid is the predominant MUFA in all nuts, and linoleic acid appears to be the predominant PUFA in all edible seeds and nuts. Compared to other nuts, walnut, para nut (Brazil nut), and cashew had the highest concentrations of PUFA [29]. In the nut and seed product category, the highest atherogenic index was found for cocoa butter (AI = 0.67). Cocoa butter and mango kernel fat have many similarities, such as having high amounts of SFA [47]. Cocoa butter and mango kernel fat contain a high 18:0 (stearic acid) content compared to the other studied oils (35.10 and 39.64%, respectively), indicating their high lipid content. According to a study by Ribeiro et al. [33] South American cocoa butter has a linoleic acid content higher than 3%, while Asian cocoa butter contains less than 2.5% linoleic acid, and cocoa butter from Africa has values between these two limits. In the category of unconventional nuts, macadamia nuts provide high amounts of MUFA (55.80%), which could be associated with lowering blood cholesterol and LDL concentrations [48].

Peanut products (nut, oil, and butter) are nutrient-dense products containing high amounts of MUFA with balanced levels of PUFA and SFA, in addition to various bioactive compounds such as coenzyme Q10, which have beneficial health effects on cardiovascular health [49,50]. Despite having polyphenolic compounds, cocoa butter lipid contains approximately 60% SFA, 35% MUFA, and a very low amount of PUFA (below 5%) (AI = 0.67, IT = 3.09), showing an increased risk of cardiovascular diseases.

Ribeiro et al. [33] emphasized that even very small differences in the amount of linoleic acid in cocoa butter from different continents reflect the geographical origin. Sonwait and Ponprachanuvut, in their study on varieties of mango cultivar originating from Thailand, found high amounts of oleic acid (46.8–42.9%) SFA, including stearic acid (41.4–37.4%), and fat content was mainly influenced by climate conditions, which were characterized by high temperatures and humidity [34]. In the nut oil category, peanut [28] and pistachio oils had higher palmitic acid (16:0) content. The major fatty acids present in peanut oil included oleic acid (51.75%) and linoleic acid (28.50%).

### 4.4. Amazonian Fats and Oils

In the Amazonian fats category, the lowest atherogenic index was found for patawa oil (AI = 0.16). The group with a medium risk of cardiovascular disease included para-nut (Brazil nut) oil and pracaxi oil, both facing the upper limit of AI; however, most Amazonian oils already reached atherogenic index values above 0.24, with tucuma oil displaying higher indices of AI and TI (0.37 and 0.80, respectively). In general, the indices and risks of cardiovascular diseases are low due to the high amount of MUFA. The major fatty acid in all investigated Amazonian oils was oleic acid (C18:1 *n*-9). Tucuma kernel fat and murumuru fat, which have the highest values (AI = 14.31 and 14.60 and TI = 6.20 and 6.69), are considered high risk for cardiovascular disease.

According to Rodrigues et al. [24] the highest oil content was observed in patawa fruit (approximately 41.8%), whereas mari fruit showed the lowest oil content (31%), indicating rich sources of fat in Amazonian fruits. In their study, the Amazonian fruits buriti and patawa had a high oleic acid content (75.5–76.7% of the total fat). Pereira et al. [25] stated that due to the high amount of MUFA, patawa and pracaxi oils showed low AI and TI. Patawa oil has very good thermostability and can be used in the food industry as a frying oil due to its high concentration of UFA, mostly oleic acid. Compared to other edible nut oils, such as almond or macadamia oils, para-nut oil has a higher linoleic acid content, which has nourishing properties. Tucuma oil is very similar to coconut oil in terms of composition; therefore, it is used in industries and pharmaceuticals [25]. Bacuri fat contains a high amount of palmitic acid, which can improve human insulin sensitivity [51].

### 4.5. Tropical Fats and Oils

Most tropical fats are solid as a result of the high SFA content. Among the abundant plant oil (placed in the category of tropical fats and oils), palm oil and palm kernel oil are high in saturated fatty acids, making them resistant to oxidation at the high temperatures used during the frying process. Interestingly, palm oil contained equal amounts of SFA and UFA. Palm oil is frequently used in the food industry due to its high oxidative stability, low formation of free fatty acids, and fatty acid composition [1]. However, since it has a high amount of palmitic acid (around 44%), its adverse health effects on blood cholesterol (increasing LDL-cholesterol) and cardiovascular risk need to be considered [52]. However, Ong and Goh confirmed no adverse health effects in connection with blood parameters in cardiovascular diseases [53]. Compared to the common oils, tropical oil has a higher amount of SFA and high indices of AI and TI (0.3–2.6), whereas common oils showed lower indices (AI = 0.1–0.2, TI = 0.1–0.4).

### 4.6. Chia Seed Oil

In addition to seafood, chia seed oil is a valuable source of essential omega-3 fatty acids and can be considered as a dietary supplement. Cold-pressed chia seed oil has an *n*-3 FA content around 65%, higher than that of linseed (55.20%), rapeseed (10.40%), and soybean (8.03%) oils. Accordingly, pressed and solvent-extracted chia seed oil has high *n*-3/*n*-6 FA (3.46–3.52) compared to the nut oils originating from temperate regions ranging from 0.01–0.22 and common vegetable oil from 0.01–0.16, indicating the beneficial health effects of chia seed. According to Ozcan et al. [26] the FA composition of chia seed oil is highly influenced by roasting and extraction methods as the heating process leads to the formation of free fatty acids and causes the oxidation of oils. This negatively affects the fatty acid composition, particularly its high PUFA content. Cultivation conditions, including climatic factors and differences in cultivation localities, might affect the FA composition of seed oil [26].

## 5. Conclusions

This study presents a view of the nutritional characteristics, particularly the FA composition, of different categories of traditional and non-traditional oils. To avoid the unhealthy effects of fats and oils, a balanced diet should include a considerable amount of vegetables and fruits, which are good sources of antioxidants. The results of the indices of AI and TI, as well as the amounts of MUFAs and PUFAs, allowed us to determine the health effects of various plant oils, nuts and seed oils, Amazonian and tropical fats and oils, and their oxidative stability attributes and formation of off-flavor compounds during the heating process, indicating their pros and cons. Mediterranean oils, including olive and avocado, which are rich in MUFA content, present higher oxidative stability compared to the other conventional and unconventional oils and can provide numerous health benefits associated with lowering the risks of chronic diseases and metabolic syndromes. Most of the common and unconventional oils originating from temperate regions are good sources of PUFA and are subsequently more prone to oxidation. Amazonian and tropical fats and oils contain high amounts of SFA, indicating higher values of AI and TI.

## Figures and Tables

**Figure 1 nutrients-14-03795-f001:**
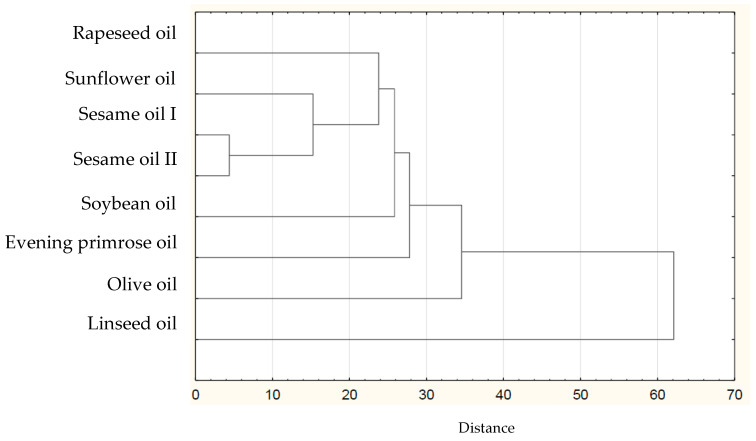
Hierarchical cluster graph of the commonly used oils. Sesame oil I - source is ref [17]. Sesame oil II—source is ref [19].

**Figure 2 nutrients-14-03795-f002:**
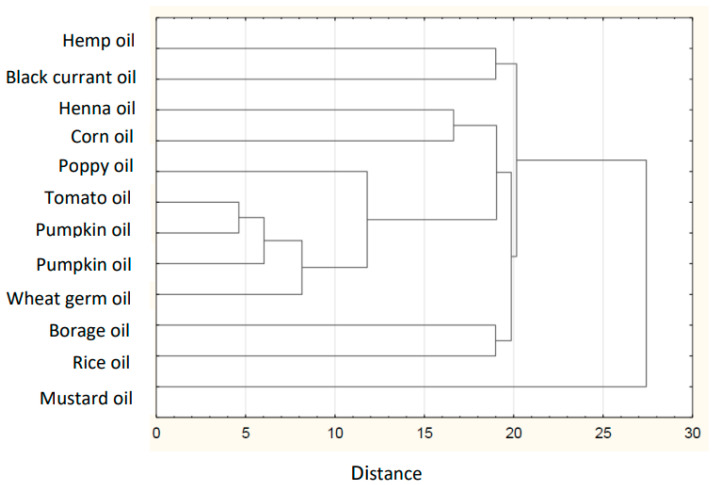
Hierarchical cluster graph of unconventional oils.

**Figure 3 nutrients-14-03795-f003:**
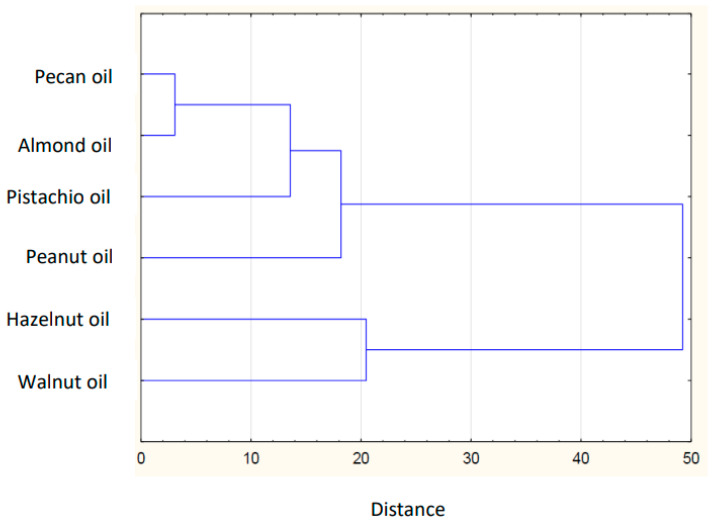
Hierarchical cluster graph of nut oils.

**Figure 4 nutrients-14-03795-f004:**
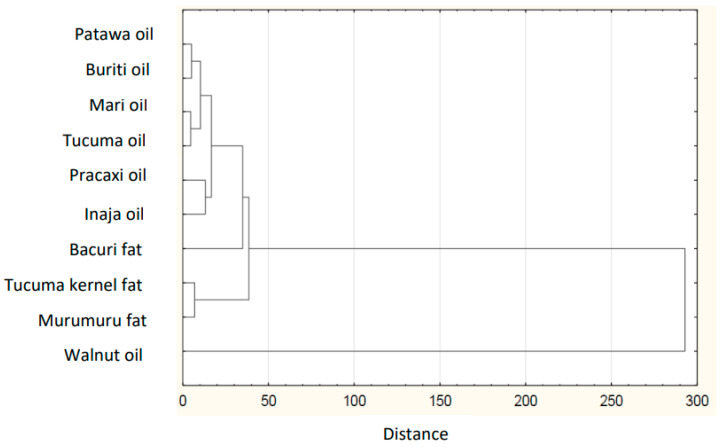
Hierarchical cluster graph of Amazonian fats and oils.

**Figure 5 nutrients-14-03795-f005:**
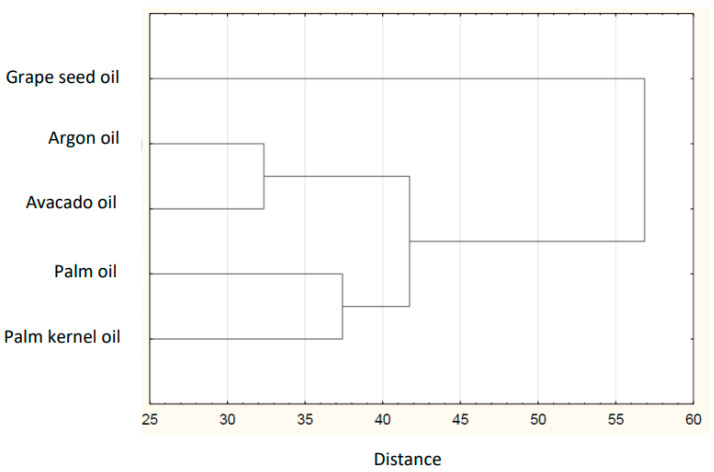
Hierarchical cluster graph of tropical fats and oils.

**Figure 6 nutrients-14-03795-f006:**
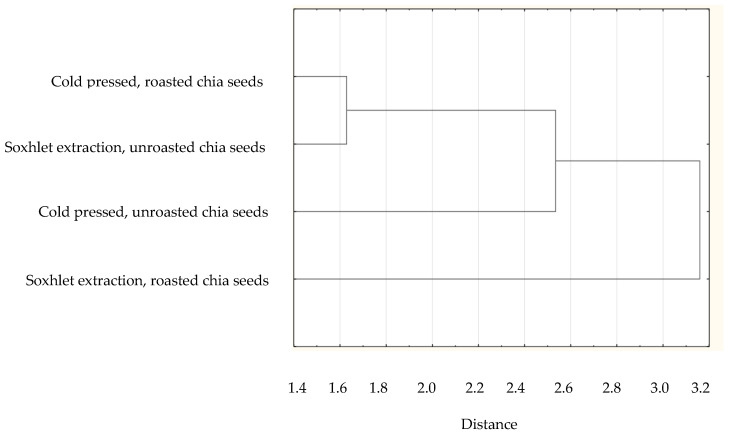
Hierarchical cluster graph of chia seed (extracted, processed and un-processed).

**Figure 7 nutrients-14-03795-f007:**
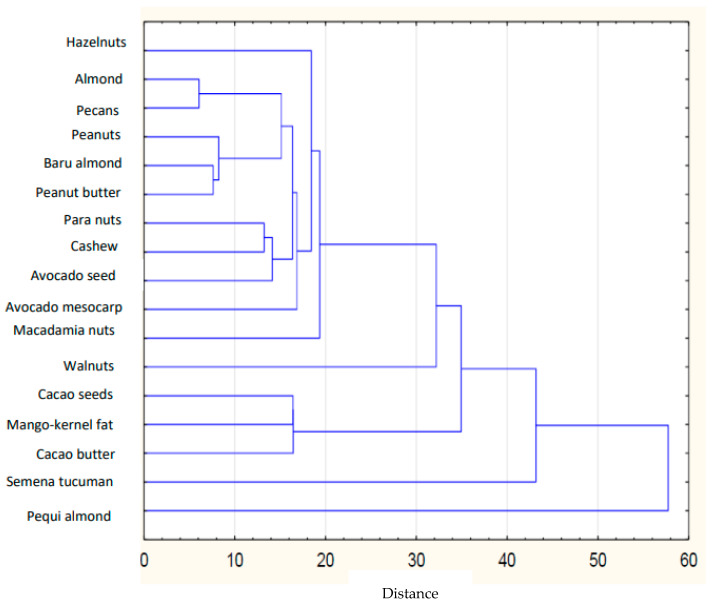
Hierarchical cluster graph of traditional nuts.

**Table 1 nutrients-14-03795-t001:** Sum of fatty acids in different types of oils, and the calculated values of atherogenic and thrombogenic indices.

Common Oil	Σ SFA	Σ MUFA	Σ PUFA	Σ *n*-3 PUFA	Σ *n*-6 PUFA	Σ *n*-3/*n*-6	AI	TI
Rapeseed oil	6.80	61.50	30.10	10.40	19.70	0.53	0.05	0.09 ^bc^
Sunflower oil	8.65	53.20	38.10	0.28	37.80	0.01	0.06	0.18 ^bc^
Linseed oil	9.20	20.60	70.20	55.20	15.00	3.68	0.06	0.05 ^bc^
Evening primrose oil	11.02	8.74	72.70	0.20	72.50	˂0.01	0.10	0.26 ^bc^
Soybean oil	13.50	24.50	62.00	8.03	54.00	0.15	0.11	0.21 ^bc^
Sesame oil	15.75	39.60	45.00	0.30	44.70	0.01	0.11	0.34 ^bc^
Sesame oil *	16.40	41.40	42.30	0.51	41.80	0.01	0.12	0.37 ^bc^
Olive oil	17.50	76.30	6.28	0.78	5.50	0.14	0.16	0.39 ^bc^
**Unconventional oils originated from the temperate region (except for Henna oil)**								
Hemp oil	9.40	13.90	76.70	18.20	56.94	0.32	0.07	0.05 ^bc^
Mustard oil	6.60	24.75	29.00	12.00	17.00	0.71	0.08	0.07 ^bc^
Blackcurrant oil	9.00	12.20	61.00	13.50	47.50	0.28	0.10	0.12 ^bc^
Henna oil	14.38	10.67	74.91	0.25	74.06	˂0.01	0.10	0.29 ^bc^
Poppy seed oil	13.50	19.70	62.90	0.50	62.40	0.01	0.13	0.32 ^bc^
Borage oil	14.30	21.85	38.50	0.20	38.30	0.01	0.18	0.46 ^bc^
Tomato oil	20.30	20.80	57.00	2.00	55.00	0.04	0.19	0.44 ^bc^
Field pumpkin (squash) oil	22.17	25.92	51.06	0.18	50.88	˂0.01	0.20	0.56 ^bc^
Pumpkin oil	20.20	24.70	55.10	0.43	54.70	0.01	0.21	0.47 ^bc^
Corn oil	21.10	0.80	67.10	1.50	65.60	0.02	0.27	0.53 ^bc^
Rice oil	18.50	39.30	35.00	1.60	33.40	0.05	0.27	0.47 ^bc^
Wheat germ oil	26.10	16.20	57.70	4.68	53.00	˂0.01	0.29	0.49 ^bc^
**Nut oils originated from temperate region**								
Pecan oil	8.00	65.20	25.90	1.10	24.80	0.04	0.07	0.17 ^bc^
Almond oil	8.79	67.10	24.00	0.14	23.90	0.01	0.07	0.19 ^bc^
Hazelnut oil	7.30	78.00	10.10	ND	ND	ND	0.06	0.18 ^bc^
Walnut oil	13.30	26.60	60.40	0.05	60.40	˂0.01	0.10	0.35 ^bc^
Peanut oil	20.80	53.15	28.60	0.10	28.50	˂0.01	0.14	0.35 ^bc^
Pistachio oil	13.80	70.40	15.50	0.30	15.20	0.02	0.15	0.30 ^bc^
**Amazonian fats and oils**								
Patawa oil	18.20	77.4	4.00	0.10	3.90	0.03	0.16	0.42 ^a^
Para nut oil (Brazil nut) oil	26.32	41.96	31.73	0.10	31.73	˂0.01	0.20	0.42 ^a^
Pracaxi oil	38.47	48.39	13.15	1.07	12.08	11.29	0.23	0.70 ^a^
Buriti oil	22.00	75.75	2.25	0.10	2.15	0.05	0.25	0.51 ^a^
Mari oil	28.40	67.90	3.50	0.10	3.40	0.03	0.29	0.59 ^a^
Tucuma oil	32.00	65.20	2.80	0.20	2.60	0.08	0.37	0.80 ^a^
Inaja oil	38.10	52.50	9.10	0.20	8.90	0.02	0.62	1.00 ^a^
Bacuri fat	64.39	33.53	2.09	0.10	2.09	20.90	1.85	3.57 ^a^
Tucuma kernel fat	88.97	8.10	2.94	0.10	2.94	29.40	14.31	6.20 ^a^
Murumuru fat	88.41	7.97	3.62	0.10	3.62	36.20	14.60	6.69 ^a^
**Tropical fats and oils**								
Grape seed oil	12.70	17.70	69.60	0.33	69.30	˂0.01	0.28	0.28 ^abc^
Argan oil	19.20	45.80	35.00	0.35	34.60	0.01	0.45	0.45 ^abc^
Avocado oil	19.20	65.30	15.50	0.73	17.70	0.04	0.41	0.41 ^abc^
Palm oil	51.20	40.60	9.75	0.50	9.25	0.05	1.88	1.88 ^abc^
Palm kernel oil	76.40	15.50	2.60	0.35	2.25	0.16	2.62	2.62 ^abc^
**Chia seed oil**								
Cold pressedRoasted chia seeds	8.69	8.15	82.23	64.98	17.25	3.77	0.07	0.04 ^c^
Soxhlet extraction (solvent extraction) Unroasted chia seeds	8.92	8.57	85.72	66.75	18.97	3.52	0.07	0.04 ^c^
Roasted chia seeds	9.55	9.11	84.58	66.24	18.34	3.61	0.08	0.04 ^c^
Cold pressed Unroasted chia seeds	9.83	9.57	87.45	67.84	19.61	3.46	0.08	0.04 ^c^
**Traditional nuts originated from temperate region**								
Hazelnuts	7.20	77.70	9.50	0.20	9.30	0.02	0.07	0.16 ^bc^
Walnuts	6.20	21.00	65.00	5.80	59.20	0.10	0.09	0.11 ^bc^
Almond	9.10	63.70	20.80	0.70	20.10	0.30	0.11	0.21 ^bc^
Peanuts	20.03	47.26	32.87	0.06	32.80	˂0.01	0.14	0.35 ^bc^
**Unconventional nuts, seeds and fruits**								
Pecans	7.60	60.90	24.60	1.00	23.60	0.04	0.12	0.17 ^abc^
Baru almond	15.47	51.57	31.71	3.14	28.57	0.11	0.13	0.24 ^abc^
*Capuassu cacao* seeds	42.90	42.28	14.80	11.00	3.80	2.89	0.14	0.72 ^abc^
Macadamia nuts	12.80	55.80	1.80	0.10	1.70	0.06	0.19	0.44 ^abc^
Avocado	20.80	65.80	12.40	0.80	11.60	0.07	0.26	0.50 ^abc^
Para nuts	25.67	28.45	45.48	0.10	45.48	˂0.01	0.26	0.68 ^abc^
Avocado seed	22.10	34.20	40.60	5.30	35.30	0.15	0.30	0.39 ^abc^
Cashew	28.67	23.03	47.99	0.47	34.07	0.01	0.40	0.95 ^abc^
Pequi almond	36.14	45.01	10.68	5.97	0.10	59.70	0.66	0.35 ^abc^
Tucuman seeds	91.36	6.45	2.13	0.10	2.10	0.05	16.29	4.95 ^abc^
**Nut and seed products**								
Mango kernel fat	49.40	44.16	4.63	0.45	4.18	0.11	0.16	1.84 ^ab^
Peanut butter	16.80	46.70	27.10	0.20	26.90	0.01	0.17	0.45 ^ab^
Cocoa butter	62.40	34.30	3.40	0.20	3.20	0.06	0.67	3.09 ^ab^

Data are presented as mean. Uncertainty of measurement is available in the reference sources. Individual FA composition sources are given in Table 2. * Data obtained from different sources. Different letters indicated significant differences (*p* ≤ 0.05) for the respective parameter among different types of oils.

**Table 2 nutrients-14-03795-t002:** Fatty acid composition of different types of seeds and fruits oil, and nuts oils.

Common Oils	C12:0	C14:0	C16:0	C16:1	C17:0	C18:0	C18:1 *n*-9	C18:2 *n*-6	C20:0	C18:3 *n*-3	C20:1 *n*-9	C22:0	C24:0	Reference
Sunflower oil	ND	0.05	4.98	0.10	0.04	3.24	53.11	37.80	0.10	0.28	0.04	0.03	0.22	[17]
Linseed oil	ND	ND	5.88	0.03	0.02	3.10	20.50	15.00	0.06	55.20	0.04	0.05	0.10	[17]
Sesame oil	ND	ND	9.82	0.11	0.02	5.96	41.12	41.80	0.30	0.51	0.12	0.10	0.18	[17]
Olive oil	ND	ND	13.60	0.50	ND	3.28	75.40	5.50	0.55	0.78	0.42	ND	0.03	[17]
Evening primrose oil	ND	0.07	8.00	0.04	ND	2.50	8.50	72.50	0.30	0.20	0.20	0.10	0.05	[18]
Sesame oil	ND	0.10	9.05	0.20	0.20	5.35	39.10	44.70	0.45	0.30	0.30	0.30	0.30	[19]
Soybean oil	ND	0.06	9.17	0.08	0.08	3.81	24.44	54.00	0.05	8.03	ND	0.05	0.24	[17]
Rapeseed oil	ND	ND	4.40	ND	ND	2.00	60.30	19.70	ND	10.40	1.20	0.40	ND	[20]
**Unconventional oils**	**C12:0**	**C14:0**	**C16:0**	**C16:1**	**C17:0**	**C18:0**	**C18:1 *n*-9**	**C18:2 *n*-6**	**C20:0**	**C18:3 *n*-3**	**C20:1 *n*-9**	**C22:0**	**C24:0**	**Reference**
Wheat germ oil	ND	0.12	21.10	0.12	0.09	2.79	15.48	53.00	1.73	4.68	0.24	0.04	0.20	[17]
Borage oil	ND	0.10	10.20	0.40	ND	3.80	17.95	38.30	0.20	0.20	3.50	ND	ND	[18]
Blackcurrant oil	ND	0.10	7.00	0.20	ND	1.50	11.00	47.50	0.20	13.50	1.00	0.10	0.10	[18]
Tomato oil	ND	0.20	13.75	0.30	0.30	5.25	20.50	55.00	0.60	2.00	ND	0.20	ND	[19]
Mustard oil	ND	0.50	2.50	0.25	ND	1.25	15.50	17.00	0.75	12.00	9.00	1.35	0.25	[18]
Rice bran oil	ND	0.70	16.90	0.20	ND	1.60	39.10	33.40	ND	1.60	ND	ND	ND	[19]
Poppy seed oil	ND	ND	10.60	ND	ND	2.90	19.70	62.40	ND	0.50	ND	ND	ND	[19]
Pumpkin oil	ND	0.22	15.60	0.15	ND	3.38	24.52	54.70	0.64	0.43	ND	0.17	0.16	[17]
Hemp oil (Order)	ND	0.04	6.08	0.10	ND	2.34	13.38	56.94	0.61	18.20	0.32	0.22	0.08	[21]
Corn oil	0.30	0.30	16.50	0.40	ND	3.30	ND	65.60	0.70	1.50	0.40	ND	ND	[19]
Henna oil	ND	˂0.01	8.47	0.10	0.10	3.86	10.19	74.06	1.68	0.25	0.38	ND	ND	[22]
Pumpkin oil	ND	0.23	14.83	0.02	ND	6.68	25.82	50.88	0.43	0.18	0.09	˂0.01	ND	[23]
**Nut oils**	**C12:0**	**C14:0**	**C16:0**	**C16:1**	**C17:0**	**C18:0**	**C18:1 *n*-9**	**C18:2 *n*-6**	**C20:0**	**C18:3 *n*-3**	**C20:1 *n*-9**	**C22:0**	**C24:0**	**Reference**
Almond oil	ND	0.02	6.73	0.48	0.05	1.78	66.69	23.90	0.11	0.14	ND	0.07	0.03	[17]
Walnut oil	ND	0.13	8.02	0.23	0.18	4.64	26.31	60.40	0.02	0.05	0.04	0.03	0.01	[17]
Hazelnut oil	ND	0.10	5.20	0.20	ND	2.00	77.80	10.10	ND	ND	ND	ND	ND	[19]
Peanut oil	0.10	0.10	11.15	0.20	ND	3.15	51.75	28.50	1.40	0.10	1.20	3.25	1.65	[19]
Pecan oil	ND	ND	6.40	0.50	ND	1.60	64.00	24.80	ND	1.10	0.70	ND	ND	[19]
Pistachio oil	ND	0.30	11.60	0.60	ND	1.40	69.30	15.20	0.50	0.30	0.50	ND	ND	[19]
**Amazon fats and oils**	**C12:0**	**C14:0**	**C16:0**	**C16:1**	**C17:0**	**C18:0**	**C18:1 *n*-9**	**C18:2 *n*-6**	**C20:0**	**C18:3 *n*-3**	**C20:1 *n*-9**	**C22:0**	**C24:0**	**Reference**
Buriti oil	0.10	0.10	18.75	0.25	0.05	1.35	75.50	2.15	1.65	0.10	ND	ND	ND	[24]
Tucuma oil	ND	0.10	24.60	0.10	0.10	3.00	65.10	2.60	4.10	0.20	ND	0.10	ND	[24]
Pataua oil	ND	0.10	13.30	0.70	0.10	4.10	76.70	3.90	0.60	0.10	ND	ND	ND	[24]
Inaja oil	3.70	7.60	20.10	0.10	ND	3.50	52.40	8.90	3.20	0.20	ND	ND	ND	[24]
Mari oil	ND	ND	20.80	0.30	0.10	6.40	67.60	3.40	1.10	0.10	ND	ND	ND	[24]
Pracaxi oil	1.20	0.71	1.95	ND	ND	2.92	47.57	12.08	1.34	1.07	ND	17.88	12.49	[25]
Bacuri fat	0.88	0.91	61.26	7.12	ND	1.33	26.41	2.09	ND	ND	ND	ND	ND	[25]
Tucuma kernel fat	50.89	25.20	6.23	ND	ND	2.74	8.10	2.94	ND	ND	ND	ND	ND	[25]
Murumuru fat	47.15	28.75	7.09	ND	ND	2.92	7.97	3.62	ND	ND	ND	ND	ND	[25]
Para nut oil (Brazil nut oil)	0.59	0.41	14.84	0.35	ND	10.48	41.62	31.73	ND	ND	ND	ND	ND	[25]
**Tropical fats and oils**	**C12:0**	**C14:0**	**C16:0**	**C16:1**	**C17:0**	**C18:0**	**C18:1 *n*-9**	**C18:2 *n*-6**	**C20:0**	**C18:3 *n*-3**	**C20:1 *n*-9**	**C22:0**	**C24:0**	**Reference**
Avocado oil	ND	0.14	16.30	4.59	0.34	1.50	60.61	14.70	0.36	0.73	0.09	0.11	0.50	[17]
Palm kernel fat	50.00	16.00	8.25	ND	ND	2.15	15.50	2.25	ND	0.35	ND	ND	ND	[19]
Palm oil	0.40	1.25	43.80	0.60	ND	4.75	40.00	9.25	1.00	0.50	ND	ND	ND	[19]
Grape seed oil	ND	0.05	7.66	0.03	0.11	4.53	17.65	69.30	0.16	0.33	0.02	0.04	0.13	[17]
Argan oil	ND	0.12	12.70	0.08	0.02	5.83	45.59	34.60	0.40	0.35	0.17	0.05	0.07	[17]
	**C12:0**	**C14:0**	**C16:0**	**C16:1**	**C17:0**	**C18:0**	**C18:1 *n*-9**	**C18:2 *n*-6**	**C20:0**	**C18:3 *n*-3**	**C20:1 *n*-9**	**C22:0**	**C24:0**	**Reference**
Roasted chia seeds—cold pressed	ND	0.04	7.19	0.05	ND	2.17	9.06	18.34	0.15	66.24	ND	ND	ND	[26]
Unroasted chia seeds—cold pressed	ND	0.05	7.28	0.09	ND	2.33	9.48	19.6	0.17	67.84	ND	ND	ND	[26]
Roasted chia seeds—soxhlet extraction	ND	0.03	6.58	0.03	ND	1.97	8.12	17.25	0.11	64.98	ND	ND	ND	[26]
Unroasted chia seeds—soxhlet extraction	ND	0.04	6.67	0.06	ND	2.07	8.51	18.97	0.14	66.75	ND	ND	ND	[26]
**Traditional nuts**	**C12:0**	**C14:0**	**C16:0**	**C16:1**	**C17:0**	**C18:0**	**C18:1 *n*-9**	**C18:2 *n*-6**	**C20:0**	**C18:3 *n*-3**	**C20:1 *n*-9**	**C22:0**	**C24:0**	**Reference**
Almond	˂0.01	0.60	6.60	ND	ND	1.90	63.70	20.10	ND	0.70	ND	ND	ND	[27]
Hazelnuts	˂0.01	0.20	5.00	ND	ND	2.00	77.70	9.30	ND	0.20	ND	ND	ND	[27]
Peanuts	ND	0.10	11.08	0.39	0.08	2.88	45.69	32.80	1.41	0.06	1.17	2.89	1.58	[28]
Walnuts	˂0.01	˂0.01	3.70	ND	ND	2.50	21.00	59.20	ND	5.80	ND	ND	ND	[27]
**Unconventional nuts, seeds and fruits**	**C12:0**	**C14:0**	**C16:0**	**C16:1**	**C17:0**	**C18:0**	**C18:1 *n*-9**	**C18:2 *n*-6**	**C20:0**	**C18:3 *n*-3**	**C20:1 *n*-9**	**C22:0**	**C24:0**	**Reference**
Cashew	ND	0.65	20.52	1.23	ND	7.16	20.76	34.07	0.10	0.47	0.15	0.17	0.15	[29]
Macadamia nuts	˂0.01	0.60	8.50	ND	ND	3.70	55.0	1.70	ND	˂0.01	ND	ND	ND	[27]
Pecans	˂0.01	˂0.01	6.10	ND	ND	1.50	60.90	23.60	ND	1.00	ND	ND	ND	[27]
Avocado mesocarp	˂0.01	0.10	19.90	5.70	˂0.01	0.70	54.40	11.60	0.10	0.80	0.40	˂0.01	˂0.01	[30]
Avocado seed	0.60	1.00	17.80	4.60	0.30	0.90	24.00	35.30	0.70	5.30	2.00	0.40	0.40	[30]
Semena tucuman	63.30	18.50	3.30	ND	0.00	1.10	6.50	2.10	ND	˂0.00	ND	ND	ND	[31]
Cacao seeds capuassu	0.09	0.05	7.60	ND	0.14	35.00	42.30	3.80	ND	11.00	ND	ND	ND	[31]
Baru almond	ND	ND	7.16	0.11	0.06	4.79	51.45	28.57	0.86	3.14	ND	0.51	1.90	[32]
Pequi almond	ND	0.34	32.55	0.02	0.71	2.39	44.76	ND	ND	5.97	ND	ND	ND	[32]
Para nuts	ND	ND	15.60	0.59	0.07	9.74	27.86	45.48	0.18	ND	ND	0.08	ND	[32]
**Nut and seed products**	**C12:0**	**C14:0**	**C16:0**	**C16:1**	**C17:0**	**C18:0**	**C18:1 *n*-9**	**C18:2 *n*-6**	**C20:0**	**C18:3 *n*-3**	**C20:1 *n*-9**	**C22:0**	**C24:0**	**Reference**
Peanut butter	˂0.01	0.40	11.10	ND	ND	5.30	46.70	26.90	ND	0.20	ND	ND	ND	[27]
Cocoa butter	ND	0.10	24.80	0.30	0.30	35.10	33.90	3.20	1.80	0.20	0.10	0.30	0.10	[33]
Mango kernel fat	0.02	0.07	7.41	ND	ND	39.64	44.16	4.18	1.57	0.45	ND	0.29	0.40	[34]

Data are presented as mean. Uncertainty of measurement is available in the reference sources. ND: not detected.

## Data Availability

Data which were used in the current study, are available in their reference article, and the calculated indices values are available from the corresponding author upon reasonable request.

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
