# Peer review of "Assessment of the Nutritional Quality of Plant Lipids Using Atherogenicity and Thrombogenicity Indices"

_nutrients, 2022, doi:10.3390/nu14183795_

Round 1

Reviewer 1 Report

Article entitled „Assessment of the nutritional quality of plant lipids using atherogenicity and thrombogenicity indices”  is pretty well written. Albeit the following changes should be done before publishing.

Introduction contain most important information about the role of n-3 fatty acids in prevention of non-communicable diseases, but authors did not give information about recommendation of n-3 and n-6 intake and the most important part which is missed is the ratio of n-3 and n-6 in diet. Based on various sources (WHO; ISSFAL or various countries in Europe, Asia USA the recommendation for selected polyunsaturated fatty acids for various age  of groups of population  (including infant and children) should be add. The authors generally have written that “…In today’s Western diet, excessive intake of n-6 PUFAs results in a very high ratio of n-6/n-3,….” Please give examples for intake and ratio of mentioned above fatty acids in various countries by various groups.

 Material and Methods

The information how the articles where selected for the showing the fatty acids profile should be add. In opinion of reviewer generally, only two  articles for each oil is not enough. Authors should add more data for example from USDA source and other governmental authorities from various countries. Additionally the range of each fatty acid should be shown in tables not only mean.

Results and discussion section should be also improved authors should underline the problem of deficiency of  various fatty acids in various countries. Based on the cited data authors should calculated amount of various oils which should be eat to improve the intake of the  deficient fatty acids. In opinion of reviewer the language should be improved by native speaker. For example more common is use "canola oil" instead of "rapeseed oil".

Author Response

Answer to the Reviewers Comments

Dear Editor and Reviewers,

We are grateful that we can submit a revised version of our manuscript. We hope that the manuscript will be suitable for publication in the “Journal of Nutrients” now. All changes are indicated in the text and reference (in the track changed version). Some new references are also added to the text and the reference list (in the track changed format). Please, find the detailed changes described below. We appreciate the time you spend to process our manuscript and looking forward to receiving your final positive decision.

Yours Sincerely,

Sarvenaz Khalili Tilami on behalf of the authors

Manuscript ID: nutrients-1892136

Assessment of the nutritional quality of plant lipids using atherogenicity and thrombogenicity indices

Raised questions

Editor and Reviewer comments:

Reviewer 1 comment’s: Introduction contain most important information about the role of n-3 fatty acids in prevention of non-communicable diseases, but authors did not give information about recommendation of n-3 and n-6 intake and the most important part which is missed is the ratio of n-3 and n-6 in diet. Based on various sources (WHO; ISSFAL or various countries in Europe, Asia USA the recommendation for selected polyunsaturated fatty acids for various age of groups of population (including infant and children) should be add.

Answer:

Thank you for your comment. With regards to the opposite properties of n-6 and n-3 PUFA and their metabolic competition (Schmitz and Ecker, 2008; Palmquist, 2009; Khalili Tilami and Sampels, 2018), it is estimated that n-6 PUFA has higher intake in the present diet. The ratio of n-6 to n-3 PUFA was around 1 during human evolution while in today’s Western diets, this ratio is 15 to 20 (Simopoulos, 2002; Khalili Tilami and Sampels, 2018). The optimal ratio for n-6 to n-3 is recommended to be 1:1 to 4:1 (Simopoulos, 2002). According to the EFSA (2010), no specific values for the n-3/n-6 ratio was set. EPA and DHA intake of at least 250 mg per day has been recommended for adults and during the pregnancy and lactation in addition to the above-mentioned intake, 100-200 mg performed DHA have been recommended. For the infants less than 6 months and young children up to 24 months, adequate intake of 100 mg DHA have been recommended (EFSA, 2010). According to the EFSA report (2010), the intake of 2 g ALA for general population have been recommended. N-6 PUFA mainly consists of linoleic acid (LA) and to a lesser amount arachidonic acid (ARA). The intake of 10 g of n-6 LA per day for adults have been recommended. No specific recommendation has been set for ARA (EFSA, 2010).

The detailed information was discussed in my review article published in the Journal of Reviews in Fisheries Science & Aquaculture entitled: Nutritional Value of Fish: Lipids, Proteins, Vitamins, and Minerals.

The authors believe that all the detailed information about the recommendations for various age groups is available in EFSA report and several reliable sources. Therefore, to avoid rewriting of all the known information, we decided to explain briefly and based on your valuable comment, we add some parts of the above-mentioned information to the text of the manuscript in the line 54-63.

The authors generally have written that “…In today’s Western diet, excessive intake of n-6 PUFAs results in a very high ratio of n-6/n-3,….” Please give examples for intake and ratio of mentioned above fatty acids in various countries by various groups.

Answer:

In the industrialized societies due to the lower consumption of foods which has higher amount of n-3 and because of consuming foods containing more n-6 such as vegetable oils (rich in n-6) (including sunflower, soybean, safflower, corn and hempseed oil) which use for cooking as well as hydrogenated oils (margarine and shortening), and foods derived from livestock animals and poultry fed on grain than on green pasture (Mariamenatu and Mohammed Abdu 2021), the ratio of n-6/n-3 can increase (Connor, 2000). It has been discussed by Simopoulos et al. (2000) that the amount of n-6 PUFA in the diet is currently very high in the US population. It has been well-documented that n-3 intake has been associated with several health effects particularly coronary heart disease, preventive effects against Alzheimer and so on. In general, eicosanoids synthesized from n-3 PUFA have immunosuppressive properties (Calder, 2001), whereas the eicosanoids from n-6 PUFA have pro-inflammatory properties which can enhance immune reactions like fever and pain (Calder, 2001).

Estimating the ratio of n-6 to n-3 in various countries was not the aim of our paper and it is not somehow possible since plant source is not the only sources of lipids. Focusing on the animal sources of lipids was not in the aim and scopes of our paper. This study aimed to compare the traditional (commonly used oil or conventional) oils and nuts with non-traditional (less used, non-conventional) plant sources of lipids to evaluate them from a nutritional point of view concerning human health by using indices of atherogenicity (AI) and thrombogenicity (TI) (line 86-88).

Material and Methods

The information how the articles where selected for the showing the fatty acids profile should be added. In opinion of reviewer generally, only two articles for each oil is not enough. Authors should add more data for example from USDA source and other governmental authorities from various countries. Additionally, the range of each fatty acid should be shown in tables not only mean.

Answer:

Thank you for the comment. The information used were selected based on the availability of data for each plant oil. These sources were based on the literature review and we wanted to make sure that we cover only the original data from the papers– where authors should declare it. EFSA, USDA source mainly do not analyze and they take the data from the literature sources.

Values for each oil were expressed as mean (different number of samples in different replications) in the papers. It is common to use the mean value of the identified fatty acids in the table of FA composition as we did it in all our published papers. However, in the time limit of five days, it is not possible to indicate the range of all the individual FAs for all the mentioned sources of plant oils. In most of the articles the range of each FA was not indicated and only the mean value was mentioned.

Results and discussion section should be also improved authors should underline the problem of deficiency of various fatty acids in various countries. Based on the cited data authors should calculated amount of various oils which should be eat to improve the intake of the deficient fatty acids. In opinion of reviewer the language should be improved by native speaker. For example, more common is use "canola oil" instead of "rapeseed oil".

Answer: Thank you for your comment. The aim of our study was not focusing on the problem of deficiency of various FAs in various countries. The main aim was to make a big search and put together the data from many sources and to calculate the indices. A lot of papers did not have calculated the indices. We designed the study to have a comparison and provide information for nutritionists – to help them to give recommendations.

Deficiency of FAs is different in different regions and countries. It is difficult to recommend the amount of oils which should be eat to improve the FA deficiency problem. Guidelines are also set for specific population and age which also depends on the region and country.

Regarding the term “canola oil” and “rapeseed oil”, historically rapeseed oil had higher erucic acid than canola oil. Canola was bred from rapeseed cultivars by Canadian in the early 1970s. They found the lower level of erucic acid in the new cultivar which was then called Canola oil. All rapeseed oils considered for human consumption has low erucic acid nowadays. Still many authors use the term “rapeseed” and these two terms are mostly equivalent.

Regarding the language, it was checked by the editing services before submitting the manuscript. For this, please find below the certificate of proofreading and language editing of the manuscript from the Editage.

Reviewer 2 Report

The paper entitled "Assessment of the nutritional quality of plant lipids using atherogenicity and thrombogenicity indice" by Tilami and KouÅ™imská  presents a view of the nutritional characteristics, particularly the FA composition, of different categories of traditional and non-traditional oils. In my view, the paper is well written for publication in this journal. Howver, there are two suggestions: 1) In table 1 and 2, the authors should add the sources informations such as species and locations. 2) The extract methods and solvents for each oil should be added.

Author Response

Answer to the Reviewers Comments

Dear Editor and Reviewers,

We are grateful that we can submit a revised version of our manuscript. We hope that the manuscript will be suitable for publication in the “Journal of Nutrients” now. All changes are indicated in the text and reference (in the track changed version). Some new references are also added to the text and the reference list (in the track changed format). Please, find the detailed changes described below. We appreciate the time you spend to process our manuscript and looking forward to receiving your final positive decision.

Yours Sincerely,

Sarvenaz Khalili Tilami on behalf of the authors

Manuscript ID: nutrients-1892136

Assessment of the nutritional quality of plant lipids using atherogenicity and thrombogenicity indices

Raised questions

Editor and Reviewer comments:

Reviewer 2:

The paper entitled "Assessment of the nutritional quality of plant lipids using atherogenicity and thrombogenicity indices" by Tilami and KouÅ™imská presents a view of the nutritional characteristics, particularly the FA composition, of different categories of traditional and non-traditional oils. In my view, the paper is well written for publication in this journal. However, there are two suggestions: 1) In table 1 and 2, the authors should add the sources informations such as species and locations. 2) The extract methods and solvents for each oil should be added.

Answer: Thank you for the comments. In the table 1, we tried to put different types of oils in the different categories based on their region. Investigating the influence of origin and extraction method on fatty acid composition of different categories of oil was not in the aim of our study. Many parameters including the extraction method, non-polar organic solvents, localities and environmental variation can influence fatty acid composition of the plant sources of lipid but our study aimed to compare the traditional (commonly used oil or conventional) oils and nuts with non-traditional (less used, non-conventional) plant sources of lipids and investigating their lipid health indices. We believe that our study is a unique comprehensive work which contain the indices information of all the plant lipid sources.

The detailed information regarding the extraction methods and solvents are available in their main sources. Together with the data, we put the corresponding reference sources which are citied. If we indicate all the detailed information, it would enlarge the paper.

But most of the oils were extracted by hexane. For instance, hexane was used for sunflower, linseed, sesame, olive, soybean, wheat germ, almond, walnut, avocado, grapeseed, argan, avocado seed and hemp oil. For mango kernel fat, soxhlet extraction method with hexane was used.
